# Metformin Prevents Key Mechanisms of Obesity-Related Complications in Visceral White Adipose Tissue of Obese Pregnant Mice

**DOI:** 10.3390/nu14112288

**Published:** 2022-05-30

**Authors:** Katrin Schmitz, Eva-Maria Turnwald, Tobias Kretschmer, Ruth Janoschek, Inga Bae-Gartz, Kathrin Voßbrecher, Merlin D. Kammerer, Angela Köninger, Alexandra Gellhaus, Marion Handwerk, Maria Wohlfarth, Dirk Gründemann, Eva Hucklenbruch-Rother, Jörg Dötsch, Sarah Appel

**Affiliations:** 1Department of Pediatrics, Faculty of Medicine and University Hospital Cologne, University of Cologne, Robert-Koch-Str. 16, 50931 Cologne, Germany; schmitzkat@gmx.de (K.S.); eva-maria.turnwald@uk-koeln.de (E.-M.T.); tobias.kretschmer@ufz.de (T.K.); ruth.janoschek@uk-koeln.de (R.J.); inga.bae-gartz@uk-koeln.de (I.B.-G.); kathrin.vr@web.de (K.V.); merlin@hokas.de (M.D.K.); privat@marionhandwerk.de (M.H.); maria.wohlfarth@uk-koeln.de (M.W.); eva.rother@uni-koeln.de (E.H.-R.); joerg.doetsch@uk-koeln.de (J.D.); 2UFZ-Helmholtz Centre for Environmental Research, Department Environmental Immunology, Permoserstraße 15, 04318 Leipzig, Germany; 3Department of Obstetrics and Gynecology, University of Regensburg, St. Hedwigs Clinic of the Order of St. John, Steinmetzstrasse 1-3, 93049 Regensburg, Germany; angela.koeninger@barmherzige-regensburg.de; 4Department of Gynecology and Obstetrics, University of Duisburg-Essen, Hufelandstrasse 55, 45122 Essen, Germany; alexandra.gellhaus@uk-essen.de; 5Department of Pharmacology, Faculty of Medicine and University Hospital Cologne, University of Cologne, Gleueler Straße 24, 50931 Cologne, Germany; dirk.gruendemann@uk-koeln.de

**Keywords:** metformin, maternal obesity, pregnancy complications, white adipose tissue, adipokines, inflammation, oxidative stress

## Abstract

With the gaining prevalence of obesity, related risks during pregnancy are rising. Inflammation and oxidative stress are considered key mechanisms arising in white adipose tissue (WAT) sparking obesity-associated complications and diseases. The established anti-diabetic drug metformin reduces both on a systemic level, but only little is known about such effects on WAT. Because inhibiting these mechanisms in WAT might prevent obesity-related adverse effects, we investigated metformin treatment during pregnancy using a mouse model of diet-induced maternal obesity. After mating, obese mice were randomised to metformin administration. On gestational day G15.5, phenotypic data were collected and perigonadal WAT (pgWAT) morphology and proteome were examined. Metformin treatment reduced weight gain and visceral fat accumulation. We detected downregulation of perilipin-1 as a correlate and observed indications of recovering respiratory capacity and adipocyte metabolism under metformin treatment. By regulating four newly discovered potential adipokines (alpha-1 antitrypsin, Apoa4, Lrg1 and Selenbp1), metformin could mediate anti-diabetic, anti-inflammatory and oxidative stress-modulating effects on local and systemic levels. Our study provides an insight into obesity-specific proteome alterations and shows novel modulating effects of metformin in pgWAT of obese dams. Accordingly, metformin therapy appears suitable to prevent some of obesity’s key mechanisms in WAT.

## 1. Introduction

The prevalence of overweight (BMI 25–29.9) and obesity (BMI ≥ 30) is rising dramatically worldwide [1]. Women of reproductive age are also affected, as epidemiologic studies from the USA and Europe recorded that about 25% of pregnant women are overweight and 7–25% are obese [2,3]. Increased body weight is a critical risk factor for adverse events during pregnancy and might have long-term negative consequences for the health of a pregnant woman and her unborn child. These include an elevated risk of gestational diabetes, hypertensive disorders, development of preeclampsia [4,5], miscarriages and stillbirths [6] as well as a lifelong increased susceptibility for metabolic and hypertensive disorders of the newborn [7]. Given this serious public health problem, the search for appropriate therapies in obese pregnancies gains more and more importance.

White adipose tissue (WAT) could be a promising target for such treatment strategies since it serves as a link between obesity and its related diseases [8]. Besides its fat storage function, WAT acts as an endocrine organ that secretes a variety of bioactive factors called adipokines [9]. Under physiological conditions, adipokines support whole-body homeostasis, while in an obese state there is a shift toward an unfavourable secretion pattern, which has proinflammatory, diabetogenic and atherogenic effects [10]. Proinflammatory adipokines also induce the production of reactive oxygen species, which leads to increased oxidative stress [11]. Obesity-associated inflammation and oxidative stress appear to originate and amplify in WAT [12,13] spreading to the entire body and causing related complications and diseases [11,14]. Together with an overall WAT dysfunction [8], inflammation and oxidative stress are considered some of obesity’s key mechanisms. Obesity-related pregnancy complications are associated with inflammatory and oxidative activities as well [15,16]. Inhibiting these processes in WAT could therefore prevent short-term and persistent damage to mother and child.

The well-established anti-diabetic drug metformin has been shown to reduce inflammation and oxidative stress on a systemic level [17,18,19]. It is able to reduce cardiac events and can exhibit positive effects on body weight [20]. Metformin has been a first-choice drug in the treatment of type 2 diabetes mellitus for decades, but the underlying mode of action is not yet fully understood. In women with pre-existing type 2 diabetes or polycystic ovary syndrome (PCOS), metformin is already used before conception and its use is continued throughout pregnancy. After detection of gestational diabetes mellitus (GDM), metformin treatment starts usually after the first trimester and is maintained until the end of pregnancy [21]. Even though previous meta-analyses and follow-up studies have shown no adverse effects [22,23,24,25], there are still divergent recommendations for metformin’s use in pregnancy, because long-term effects on the child have not been thoroughly researched yet. So far, only little is known about metformin’s impact on maternal WAT in obese pregnancies. To shed light on this topic, we used a mouse model of maternal obesity and started metformin treatment at the onset of pregnancy. We particularly chose this timing, as for other interventions it is known that they have only little effect on e.g., maternal metabolic conditions when initiated after the first trimester. This is probably due to the fact that by the end of the first trimester, many physiological adaptions to pregnancy have already occurred [26]. Investigating the effects on visceral WAT was of primary interest, as it exhibits a great metabolic and inflammatory activity [27,28] and visceral fat mass correlates more strongly with obesity-associated diseases than obesity in general [29]. Hence, we used perigonadal WAT (pgWAT) as a large visceral fat pad around the uterus for this study. The aim was to detect possible favourable changes by metformin treatment during obese pregnancy in the dam and pgWAT phenotype. Moreover, we analysed the pgWAT proteome, with a particular focus on potential markers of metabolic and inflammatory alterations, oxidative stress and tissue dysfunction to be altered as a result of metformin treatment. 

## 2. Materials and Methods

### 2.1. Animal Model

The experimental design was approved by the local government authorities (LANUV NRW, *State Agency for Nature, Environment and Consumer Protection North Rhine-Westphalia*, Recklinghausen, Germany; 84-02.04.2016.A046) and all procedures were conducted following German animal welfare guidelines. C57BL/6N mice (Janvier Labs, Le Genest-Saint-Isle, France) were kept at the animal housing network at the University Hospital of Cologne (Cologne, Germany). After weaning at three weeks of age, mice received either a standard diet (SD; R/MH, ssniff^®^, Soest, Germany) or an obesity-inducing high-fat diet (HFD; C1057 modified, Altromin, Lage, Germany). For further details on animal housing and composition of diets see [30,31] and Appendix A (Appendix A). At the age of 12–20 weeks, SD and HFD mice were weighed and mated overnight (threshold of <23.5 g for SD and >23.0 g for HFD). The next day was defined as gestational day (G) 0.5. From G0.5 on until the end of the experiment, randomly chosen animals of the HFD group were treated with metformin (Alfa Aesar, Kandel, Germany) via drinking water (dose of approximately 432 mg/kg body weight/day). SD and HFD were continuously fed during mating and gestation period, resulting in three different intervention groups: SD and HFD groups were only provided with the respective diet while the HFD + MF group was, additionally to feeding an HFD, treated with metformin during pregnancy. On G15.5, which corresponds to the middle of the last trimester, mice were analgised by intraperitoneal injection of buprenorphine (0.1 mg/kg body weight) and euthanised 30 min later by CO_2_ inhalation as described previously [32]. A flow chart explaining the experimental setup of the mouse groups can be found in the Appendix A (Appendix A). At the time of sampling, dams had intact pregnancies with at least one living fetus. Body weight and length were determined postmortally. Blood was taken by cardiac puncture to obtain maternal serum. PgWAT was removed and weighed as a whole. One part of the pgWAT was frozen in liquid nitrogen for proteome analysis, and samples were stored at −80 °C until subsequent analysis. The other part was fixated in formaldehyde (Roti^®^-Histofix 4%, Carl Roth, Karlsruhe, Germany) and embedded in paraffin for histological analysis.

### 2.2. Histological Analysis of Fat Cell Size

The 5 µm thick histological sections of pgWAT were Hematoxylin and Eosin (Carl Roth, Karlsruhe, Germany) stained following standard procedures. Five sections representing separate areas of every pgWAT sample were digitised at 40× magnification (SCN400, Leica, Wetzlar, Germany) and cut into images of a defined size of 300 × 300 µm using QuPath image analysis software (University of Edinburgh, Edinburgh, Great Britain; version 0.2.0 m2) [33]. Five of these images were randomly selected (one image from each of the digitised sections) for automatic analysis by the ImageJ Fiji (https://fiji.sc, accessed on 11 May 2021) [34] plug-in Adiposoft (CIMA, Universidad de Navarra, Pamplona, Spain; version 1.16) [35] where fat cells were automatically marked and measured. A high correlation between manual measurements and Adiposoft results has already been shown [35]. The minimum (10 µm) and maximum diameter (140 µm) were determined by an exemplary analysis of images with very small and very large fat cells. All truncated fat cells at the edge of an image were automatically excluded. The resulting automatic marking of the fat cells in each image was then checked manually and cell markings that apparently fulfiled one of the following criteria were deleted: a mark does not contain a cell, cell boundaries cannot be clearly identified or reconstructed, a mark divides a cell into several parts or connects several cells into one or marking deviates significantly from the actual cell boundaries. Then the average area and diameter of the fat cells were normalised to the number of counted cells as follows. Fat cell sizes were analysed in a blinded manner.
$${\bar{x}}_{\mathrm{total}}\;=\;\frac{\left({\bar{x}}_{\mathrm{img}1}\;\times \;C_{\mathrm{img}1})\;+\;({\bar{x}}_{\mathrm{img}2}\;\times \;C_{\mathrm{img}2})\;+\;({\bar{x}}_{\mathrm{img}3}\;\times \;C_{\mathrm{img}3})\;+\;({\bar{x}}_{\mathrm{img}4}\;\times \;C_{\mathrm{img}4})\;+\;({\bar{x}}_{\mathrm{img}5}\;\times C_{\mathrm{img}5}\right)}{C_{\mathrm{total}}}$$

${\bar{x}}_{\mathrm{total}}$: mean area or mean diameter of fat cells in a pgWAT specimen,

${\bar{x}}_{\mathrm{img}}$: mean area or mean diameter of fat cells in one image,

$C_{\mathrm{img}}$: cell count of one image,

$C_{\mathrm{total}}$: total cell count of all five images of a pgWAT specimen.

### 2.3. Quantification of Metformin in Serum by Mass Spectrometry

Metformin levels in serum were measured using an established liquid chromatography-mass spectrometry (LC-MS) method [36]. For this purpose, serum samples were diluted at 1:100 with acetonitrile. A total of 10 µL of each sample were examined by mass spectrometry and compared to a calibration series of 0–50 ng/mL metformin.

### 2.4. Protein Precipitation, Digestion and StageTip Purification

Proteins were prepared for mass spectrometry (pgWAT from *n* = 5 dams in SD and HFD groups, *n* = 4 dams in HFD + MF group) following the protocols of CECAD (Cluster of Excellence-Cellular Stress Responses in Aging-Associated Diseases) Proteomics Facility (Cologne, Germany) [37]. The pgWAT was lysed using modified RIPA buffer (50 mM TRIS-hydrochloride, 150 mM NaCl, 1% (*v*/*v*) IGEPAL^®^, 0.25% (*v*/*v*) deoxycholic acid sodium salt, 1 mM EDTA, 1 µg/mL aprotinin, 1 µg/mL pepstatin A, 1 µg/mL leupeptin, 1 mM PMSF, 1 mM NaF, 1 mM Na_3_VO_4_) and sonicated (3–6 times, 20 s at 50% energy; SONOPULS^®^ HD 2070, Bandelin electronic, Berlin, Germany). Samples were incubated on ice for 1 h and centrifuged (5 min, 16,000× *g*, 4 °C). From the supernatant, proteins were precipitated using four times their volume of ice-cold acetone. Samples were again incubated (15 min −80 °C, 90 min −20 °C) and centrifuged (5 min, 16,000× *g*, 4 °C). The remaining protein pellet was washed twice with ice-cold acetone (5 min, 16,000× *g*, 4 °C), air-dried and resuspended in 200 µL of urea solution (8 M). Subsequently, 100 µL of urea buffer (8 M urea, 50 mM TEAB) and 2 µL of cOmplete^®^ protease inhibitors (Roche, Basel, Switzerland) were added, hereupon chromatin was degraded using a Bioruptor^®^ (10 min, cycle 30/30 s; Diagenode, Ougrée, Belgium). Another 100 µL buffer and 2 µL protease inhibitors were added to samples with a still undissolved pellet and were again sonicated for 30 s (50% of energy; SONOPULS^®^ HD 2070). After a 15 min centrifugation (20,000× *g*), protein concentrations were measured by Multiskan FC (Thermo Scientific, Waltham, MA, USA). A total of 50 µg protein from each sample was filled up with urea buffer to a final volume of 100 µL. Then, dithiothreitol was added (final concentration of 5 mM), vortexed and the sample was incubated for 1 h at 37 °C. Chloracetamide was added to a final concentration of 40 mM, vortexed and incubated for 30 min at room temperature. Proteins were digested with lys-c and trypsin with an enzyme-substrate ratio of 1:75. Finally, samples were subjected to StageTip purification (reversed phase C_18_ and polystyrene-divinylbenzene copolymer) and were stored at 4 °C until LC-MS analysis.

### 2.5. Proteomic Screen by Mass Spectrometry and Further Data Analysis

LC-MS was performed, and the resulting data were further analysed by the CECAD Proteomics Facility using MaxQuant software (Max Planck Institute of Biochemistry, Planegg, Germany; version 1.5.3.8) [38] as well as Perseus software (Max Planck Institute of Biochemistry, Planegg, Germany; version 1.6.1.1) [39], according to a previous publication from our laboratory [30]. Values had to be imputed for samples in which a protein was not detected (in a range of 1.8 ± 0.3 σ from the mean) to be able to execute statistical tests. The number of non-imputed values was given as valid values, which consequently indicates the number of samples in which an individual protein was successfully detected. Subsequent data processing with Perseus was carried out by our research group analogous to the facility’s advice. A principal component analysis was performed to analyse the clustering of the samples. Comparing HFD vs. SD and HFD + MF vs. HFD, a two-sample t-test was conducted for each detected protein. *p*-values were corrected for multiple testing using a permutation-based method [40] to reduce the number of false-positive findings. This resulted in corresponding adjusted q-values. Alterations with a fold change of at least 2^0.59^ (≈1.5-fold) and a q-value of less than 0.05 were considered significant.

In the following, only significantly changed proteins, which were either detected in all samples of a group (valid values = 5 in SD and HFD groups; valid values = 4 in HFD + MF group) or were not detected in any sample of a group (valid values = 0), were considered as relevantly altered and therefore examined in detail. Significant changes based on 0 valid values in both compared groups were excluded. To illustrate overlapping proteins between the two comparisons, the significantly altered proteins (fold change >1.5, q <0.05) were displayed in a Venn diagram using FunRich software (http://www.funrich.org; version 3.1.3, accessed on 11 May 2021) [41]. Moreover, an analysis of potential interactions between the significantly changed proteins (based on experiments and curated databases) and functional enrichments (regarding biological processes, cell components, Uniprot keywords and KEGG or Reactome pathways) was carried out using STRING Database (https://string-db.org; version 11.0, accessed on 11 May 2021) [42]. UniProt database [43] was utilised to find homologous IDs for unrecognised protein IDs by STRING.

### 2.6. Statistical Analyses

Statistical analyses were carried out using GraphPad Prism 8.3.0. Histological data on fat cell sizes were subjected to a Grubbs test, in which no significant outliers were identified (*p* = 0.05). Other data did not undergo a Grubbs test. A Shapiro–Wilk test was performed to test for normal distribution of data. One-way ANOVA with a suggested post hoc test (either Sidak’s or Tukey’s multiple comparisons test) was then used for normally distributed data when comparing more than two groups. Non-normally distributed data were compared using the Mann–Whitney test for two groups or Kruskal–Wallis test (with Dunn’s multiple comparisons test) for more than two groups. Metformin levels are given as median with the interquartile range. Phenotypic and histological data are displayed as mean ± standard deviation. 

## 3. Results

### 3.1. Metformin Serum Levels

Serum levels were determined postmortem to assess metformin treatment. There was no evidence of metformin in the serums of SD and HFD groups, as expected. The HFD + MF group exhibited median metformin levels of 2.24 (1.19 to 3.10) µg/mL.

### 3.2. Impact on Body and pgWAT Weight 

At mating (G0) (Figure 1a), HFD mice (weighing 25.81 ± 2.01 g) as well as HFD mice reserved for metformin treatment after mating (HFD+MF; weighing 24.82 ± 1.20 g) were significantly heavier than SD mice (weighing 21.68 ± 0.72 g). At G15.5 (Figure 1a), HFD mice weighed 35.76 ± 2.64 g and still displayed a significantly higher body weight than SD mice (31.68 ± 2.04 g). Metformin-treated dams (HFD + MF) exhibited a significantly lower body weight (30.64 ± 2.63 g) than HFD dams at G15.5 corresponding to a weight reduction of 14.3%. The mean weight gain during the gestation period (Figure 1b) was almost equal among SD and HFD groups with 9.82 ± 1.96 g and 9.93 ± 1.81 g respectively. The HFD + MF group gained significantly less weight (5.82 ± 2.23 g) than HFD dams, which corresponds to a reduced weight gain of 41.4% through metformin treatment during pregnancy. Relating weight to body length showed no significant effect on these results (Figure 1c). The groups’ litter sizes did not differ significantly from one another and at G15.5 dams had intact pregnancies with a median of 8 (6 to 9) fetuses each (Figure 1d).

Collected pgWAT at G15.5 weighed 0.299 ± 0.079 g in the SD group, 0.999 ± 0.330 g in the HFD group and 0.531 ± 0.228 g in the HFD + MF group. A ratio of pgWAT and body weight was calculated to normalise the measured pgWAT values (Figure 1e). In the HFD dams, a significantly higher percentage of their body weight consisted of pgWAT (2.72 ± 0.81%) than in SD mice (0.94 ± 0.23%). Following metformin treatment, the amount of pgWAT was significantly reduced to 1.72 ± 0.66% of the HFD + MF dams body weight. This corresponds to a 37% reduced pgWAT mass in HFD + MF dams compared to HFD dams on G15.5.

### 3.3. Metformin Treatment Appears to Reduce Fat Cell Size in pgWAT of Obese Dams

Metformin treatment and HFD feeding had an impact on measured fat cell sizes in histological pgWAT sections (Figure 1f). The mean area (Figure 1g) and mean diameter (Figure 1h) of fat cells in pgWAT of the HFD mice differed significantly from that in SD mice. In comparison with the HFD-fed mice, a trend toward a smaller area (*p* = 0.0849) of fat cells was seen in HFD + MF mice. While in HFD-fed mice the fat cell area in pgWAT was increased by 125% compared to SD, metformin treatment reduced fat cell areas by 20% compared to HFD (SD = 1165 ± 296.8 µm^2^, HFD = 2624 ± 701.9 µm^2^, HFD + MF = 2101 ± 643.9 µm^2^). Fat cell diameter in HFD mice was also significantly greater than in SD mice, while in HFD + MF mice there was no significant diameter reduction compared to the HFD group (SD = 36.54 ± 5.64 µm, HFD = 54.04 ± 7.31 µm, HFD + MF = 48.94 ± 7.82 µm).

### 3.4. Substantial and Differing Effects on pgWAT Proteome by HFD and Metformin Treatment

A principal component analysis of the 3577 overall detected proteins showed differing global protein expression patterns in the pgWAT of SD, HFD and HFD + MF dams (Figure 2a). Global protein expression in SD-fed dams and HFD-fed dams were distinct from each other and also differed in comparison to the expression in HFD + MF-treated dams.

By analysing overlaps, five proteins were found to be regulated in pgWAT by HFD feeding (HFD vs. SD) as well as metformin treatment of HFD-fed dams (HFD + MF vs. HFD) (Figure 2b). Of these proteins, Apoa4 (*apolipoprotein A-IV*) was upregulated in both cases, while Mlycd, Pdk2, Opa3 and Cyp2s1 (*malonyl-CoA decarboxylase, pyruvate dehydrogenase kinase isozyme 2, optic atrophy 3 protein homolog and cytochrome P450 2S1*) were counter regulated in HFD + MF group ( Figure 2c). Overall, there were more significant and relevant changes observed in the pgWAT proteome due to HFD feeding (196 altered proteins) than through metformin treatment (103 altered proteins).

### 3.5. HFD Feeding Alters the Expression of 196 Proteins in pgWAT Proteome of Pregnant Mice

By HFD feeding (HFD vs. SD), 143 proteins were significantly and relevantly downregulated while 53 were upregulated in pgWAT proteome (for detailed information see Appendix A in the Appendix A). A total of 85 potential interactions and various functional enrichments were demonstrated by STRING-analysis among these downregulated proteins (Figure 3a). A total of 54 of these were mitochondrial proteins, including proteins of all five respiratory chain complexes with a notable focus on complex I. Pyruvate carboxylase (Pcx), the catalysing enzyme for the initial reaction of de novo lipogenesis and gluconeogenesis, was downregulated. In addition, Mlycd and Pdk2 as regulatory proteins in glucose and lipid metabolism were downregulated. There was also downregulation of numerous other proteins of lipid metabolism relating to de novo lipogenesis (Acly, Fasn, Acaca, Acss2, Acss3, Acsf3; *ATP-citrate synthase,*
*fatty acid synthase, acetyl-CoA carboxylase 1, acetyl-coenzyme A synthetase, acyl-CoA synthetase short-chain family member 3, acyl-CoA synthetase family member 3*) lipogenesis regulation (Thrsp, Mid1ip1; *t**hyroid hormone-inducible hepatic protein, mid1-interacting protein 1*) and β-oxidation (Mlycd and Acat2, Acadsb, Acox1, Gcdh; *acetyl-CoA acetyltransferase, short/branched chain specific acyl-CoA dehydrogenase, peroxisomal acyl-coenzyme A oxidase 1, glutaryl-CoA dehydrogenase*). Four glycolytic proteins (Pkm, Gpi, Hk2, Pfkl; *p**yruvate kinase PKM, glucose-6-phosphate isomerase, hexokinase-2, ATP-dependent 6-phosphofructokinase*) and various proteins from multiple other metabolic pathways, such as acetyl-CoA synthesis, pyruvate and glutamate metabolism, and the pentose–phosphate pathway were also downregulated (Pdha1, Dld, Me1, Aldh4a1, Glul, Tkt; *p**yruvate dehydrogenase E1 component subunit alpha, dihydrolipoyl dehydrogenase, NADP-dependent malic enzyme, delta-1-pyrroline-5-carboxylate dehydrogenase, glutamine synthetase, transketolase*). Five proteins involved in glycogen metabolism were downregulated as well, which apply to both glycogen synthesis and glycogenolysis (Gys1, Gbe1, Ugp2, Agl, Pygl; *glycogen [starch] synthase, 1,4-alpha-glucan-branching enzyme, UTP-glucose-1-phosphate uridylyltransferase, 4-alpha-glucanotransferase, glycogen phosphorylase*). With aldehyde dehydrogenase-2 (Aldh2) and thioredoxin-2 (Txn2), two mitochondrial proteins that counteract oxidative stress were downregulated. Furthermore, two thioredoxin reductases (Txnrd1, Txnrd2; *t**hioredoxin reductase 1, thioredoxin reductase 2*) and the antioxidant Mpst (*Sulfurtransferase*) that is activated by thioredoxin were downregulated. 

Among upregulated proteins, there were 20 potential interactions identified (Figure 3b). A total of 16 of these proteins belong to the biological process of cellular component organisation or cytoskeleton organisation. Plenty of proteins were part of the cytoskeleton themselves or participated in its organisation, most of them belonging to the actin cytoskeleton (Sptan1, Sptbn1, Prkcdbp, Vim, Cald1, Itgb5, Tln2, Tpm4, Myh10, Arrb1, Anxa1; *spectrin alpha chain non-erythrocytic 1, spectrin beta chain non-erythrocytic 1, protein kinase C delta-binding protein, vimentin, caldesmon 1, integrin beta, talin-2, tropomyosin alpha-4 chain, myosin-10, beta-arrestin-1, annexin A1*). Six proteins were involved in cell adhesion processes (Sptan1, Itgb5, Tln2, Myh10 and Itga6, Gak; *i**ntegrin alpha-6, cyclin-G-associated kinase*). Laminin (Lamc1) as a component of the basal lamina and thus a component of the extracellular matrix was also upregulated.

### 3.6. Metformin Treatment Alters the Expression of 103 Proteins in pgWAT Proteome of Obese Pregnant Mice

Metformin treatment of HFD-fed dams significantly and relevantly downregulated 45 proteins and upregulated 58 proteins in the pgWAT proteome (for detailed information see Appendix A in the Appendix A). Through STRING analysis, potential interactions could be detected between 16 of the downregulated proteins (Figure 4a); however, functional enrichments were rare. Perilipin-1 (Plin1) as part of the lipid vacuole was downregulated 15.7-fold.

There were potential interactions among 18 of the upregulated proteins determined (Figure 4b). Most proteins belonged to the nucleus and the mitochondrion. Proteins from complex I (Ndufv3, Ndufb2; *NADH dehydrogenase [ubiquinone] flavoprotein 3, NADH dehydrogenase [ubiquinone] 1 beta subcomplex subunit 2*) and complex V (Atp5o; *ATP synthase subunit O*) of the respiratory chain, as well as other proteins of mitochondrial organisation (Opa3 and Etfa, Gfm1; *Electron transfer flavoprotein subunit alpha, elongation factor G*) were upregulated. With Mgst1, Mgst3 and Naprt (*microsomal glutathione S-transferase 1, microsomal glutathione S-transferase 3, nicotinate phosphoribosyltransferase*), three proteins were upregulated that reduce oxidative stress. An upregulation of proteins of the immune response (A2m, Serpina1b, Serpina1c; *alpha-2-macroglobulin-P, alpha-1-antitrypsin 1-2, alpha-1-antitrypsin 1-3*) was observed. Various proteins involved in lipid metabolism (Mlycd, Apoa4, Pdk2 and Acot13; *acyl-coenzyme A thioesterase 13*) were upregulated, as was the glycolytic Gapdh (*glyceraldehyde-3-phosphate dehydrogenase*). Three potentially secreted proteins were upregulated (Serpina1b, Serpina1c, Apoa4), while two were downregulated (Lrg1, Selenbp1; *Leucine-rich HEV glycoprotein, Selenium-binding protein 1*).

## 4. Discussion

To investigate metformin’s impact on mechanisms of obesity-related complications and diseases during pregnancy, we analysed pgWAT at the middle of the last trimester (G15.5) from SD, HFD and HFD + MF dams. Despite many years of clinical use, the therapeutic range for metformin is not clearly defined. However, therapeutic serum levels between 0.1 and 4 µg/mL are often cited in the literature [44]. During pregnancy, daily metformin intake ranges from 500 to 3000 mg per day [22,45]. After dosage of 1000 mg metformin twice daily, plasma metformin concentrations peak at about 1.5 µg/mL + 0.5 (mean + SD) [45]. The median metformin serum level of the HFD + MF mice (2.24 µg/mL) would thus be in the upper therapeutic range for non-pregnant individuals, but slightly above the therapeutic range detected in pregnant women. This has to be considered when looking at the results of this study.

Excessive weight gain during pregnancy leads to a more frequent occurrence of obesity-associated complications [46]. Hence, for overweight or obese women, weight gain during pregnancy is recommended to be lower than in lean women [47,48]. Some even recommend for obese women lose rather than gain weight during pregnancy [49]. This however is controversially discussed as a strong loss in weight gain during pregnancy might be a problem for the unborn child [50]. Weight retention after birth can induce a further progression of obesity and is associated with an increased risk of complications during the next pregnancy [3]. Because one out of two obese pregnant women is affected by excessive weight gain [51], metformin treatment could help to comply with the recommended weight gain guidelines and thus prevent complications. Our study shows that metformin treatment during gestation significantly reduced weight gain and visceral fat accumulation in obese mice. This is consistent with the lower weight gain observed in clinical trials on metformin therapy in diabetic and non-diabetic pregnant women [22,52]. It moreover supports the assumption that metformin treatment reduces visceral fat mass, as already observed in two studies on humans and rats [53,54]. In the histological analysis of pgWAT morphology, metformin treatment reduced fat cell hypertrophy in obese dams. Even though this effect was not significant, it supports the findings from Souza-Mello et al., who also found a reduced fat cell size in murine visceral WAT due to metformin treatment [55]. Visceral WAT is a crucial risk factor for obesity-associated diseases [27,28,29], and hypertrophic adipocytes are associated with reduced insulin sensitivity, increased inflammatory response as well as oxidative stress [56,57]. Accordingly, the observed beneficial effects on obese dam’s body weight due to metformin therapy are accompanied by reduced visceral adipose tissue mass and likely decreased fat cell hypertrophy. Interestingly, the notable downregulation of perilipin-1 in the pgWAT proteome of metformin-treated obese dams provides a previously unknown correlate to weight and cell size reducing effects of metformin. Perilipin-1 envelops the lipid vacuole of fat cells and is involved in the regulation of lipolysis [58]. Its absence leads to a lean phenotype with resistance to diet-induced obesity, increased adipocytic lipolysis and an increased release of the protective adipokine leptin in mice [58,59].

To better understand metformin’s impact, alterations in the pgWAT proteome of obese compared to lean dams will be initially discussed: Obesity appears to induce a fibrotic remodelling with upregulated proteins of cytoskeleton, cell adhesion and extracellular matrix to adapt to the increased mechanical stress caused by expansion of fat tissue. It is already known that adipocytes’ metabolic function becomes disturbed during this process [60], which is reflected by the extensive impairment of central metabolic pathways in pgWAT of obese dams. The corresponding changes in protein expression suggest a decreased de novo lipogenesis, reduced β-oxidation of fatty acids and impaired glucose and glycogen metabolism. The observed indications of reduced respiratory and antioxidant capacity in the HFD group can be implied as another significant change that is already known to be associated with obesity [13,61]. Altogether, these alterations in obese dam’s pgWAT proteome complete other studies that found matching metabolic changes and remodelling (at the level of mRNA, metabolic products, genes or protein phosphorylation) in WAT of obese mice [62,63,64]. 

Metformin treatment of obese dams seems to partially recover respiratory capacity and mitochondria count, as various mitochondrial proteins and parts of respiratory chain complexes were upregulated when compared to pgWAT of obese dams. In addition, some proteins of lipid metabolism and its regulators were upregulated. Mlycd and Pdk2 were directly counter regulated in obese dams receiving metformin treatment. Since both enzymes regulate the balance between lipid and glucose metabolism [65,66], their upregulation favours fatty acid utilisation over glucose. These changes could be interpreted as a beginning counteraction of obesity-associated dysregulation toward a more physiological and balanced adipocyte metabolism. As WAT dysfunction is an important driver of obesity-related diseases [10], this can be stressed as a further positive effect of metformin treatment on pgWAT.

Regarding inflammation and oxidative stress as key mechanisms of obesity-related complications, various metformin effects were observed on pgWAT of obese mice. The upregulation of several antioxidant enzymes might be a sign of enhanced oxidative capacity. A further indicator of reduced oxidative stress by metformin could be the increased expression of Gapdh, which serves as a “metabolic switch” to antagonise oxidative stress [67]. By downregulating cytochrome P450 2S1 (Cyp2s1), metformin might convey anti-inflammatory effects, as some of Cyp2s1’s products play a critical inflammatory role [68]. 

The identification of four novel potential adipokines is an essential finding, as metformin might presumably mediate systemic effects based upon their regulation in pgWAT. One of these potential adipokines is the upregulated alpha-1 antitrypsin (i.a. Serpina1b and Serpina1c) in HFD + MF compared to HFD. It exhibits anti-inflammatory as well as immunomodulatory effects [69], and it also appears to increase insulin secretion and protect β-cells in the pancreas from cytokine-induced apoptosis [70]. Alpha-1-antitrypsin deficiency is associated with type 2 diabetes [71], and therapy with alpha-1-antitrypsin is already being tested as a treatment strategy for type 1 diabetes [72]. Another potential adipokine is the upregulated Apoa4, whose plasma concentration initially rises due to a fatty diet, but falls thereupon in the long term [73]. It has various properties such as anti-inflammatory action, increasing insulin secretion and activating lipoprotein lipase [74], which is required for lipid uptake by adipocytes from circulation. An Apoa4 deficiency is associated with obesity-related diseases [74]. Its upregulation in obese compared to lean dams could be seen as an initial compensatory mechanism that is even fortified by metformin treatment. Metformin treatment of obese dams also downregulated the potential adipokine Lrg1. This protein is already being discussed as a novel biomarker for inflammation and the risk of cardiovascular diseases [75]. Based on our findings, it stands to reason that Lrg1 is involved in the systemic spread of visceral WAT inflammation. Little is known about the role of the downregulated Selenbp1 in WAT of metformin-treated obese dams compared to obese dams. Selenbp1 is an intracellular protein that is considered a marker of cell differentiation and mature adipocytes [76]. However, it appears to regulate the intra- and extracellular redox status [77] and is also discussed as a biomarker for cardiac events [78], and hence, could serve as a potential adipokine, too. Since the secretion of the proteins by WAT has not been sufficiently investigated yet, these findings should be an impulse for further research. By regulating these potential adipokines, metformin can possibly act as anti-inflammatory, anti-diabetic and oxidative stress-modulating in visceral WAT and on a systemic level.

Finally, Opa3 is a promising protein that was downregulated in obese dams and changed in the opposite direction through metformin treatment. Even though to date little is known about its function in WAT, it has been associated with regulating mitochondrial function, lipid metabolism, abdominal fat mass and thermogenesis in adipose tissues [79]. Enhancing energy expenditure in brown adipose tissue by thermogenesis is an attractive therapeutic approach for obesity [80], and an increased expression of markers for brown adipose tissue in WAT of obese mice has already been shown for metformin [81]. Thus, Opa3 could potentially mediate diverse metformin effects on pgWAT. 

Due to the complexity of a proteomic screen, we were only able to present and discuss an extract of our results. 

Proteome analysis by mass spectrometry is always limited by the fact that protein frequency in biological tissues varies by several orders of magnitude, so proteins with higher frequency can mask less frequent proteins [82]. Additionally, post-translational modifications and subsequently altered protein activity were not considered. It must also be taken into account that there is no matching fat depot to the murine pgWAT in humans, as they form intra-abdominal WAT in other locations [83]. Due to its size and accessibility, using pgWAT as a surrogate for human visceral WAT is common [83]. However, using a different fat depot like mesenteric fat might have been more appropriate as it drains its blood into the portal vein, which can be of great importance in metabolic disorders. Gaining knowledge from animal models is essential, as such research on human visceral WAT could not be legitimated in clinical studies without an appropriate indication for surgery. In addition, it should be considered that the outcome of pregnancy was not examined in this study. Metformin’s impact on pregnancy and fetal outcome in our mouse model will be discussed in independent publications.

## 5. Conclusions

In conclusion, our study showed that metformin therapy during gestation reduces weight gain and visceral WAT mass. It offers plenty of effects suitable for inhibiting some of obesities key mechanisms in visceral WAT as inflammation, oxidative stress and tissue dysfunction. Our novel findings encourage further research on the potential of metformin treatment in obese pregnant women to prevent obesity-related complications. 

## Figures and Tables

**Figure 1 nutrients-14-02288-f001:**
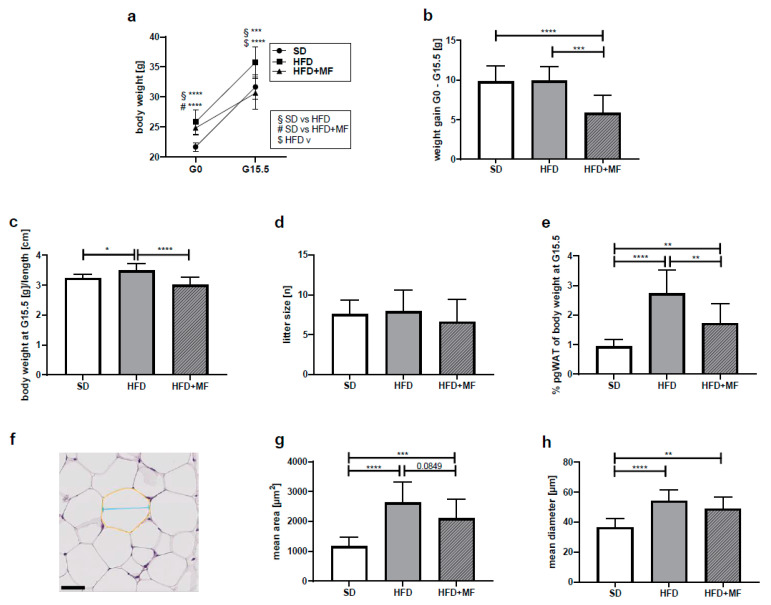
Metformin treatment reduced gestational weight gain, pgWAT mass and fat cell size in obese dams (**a**) Body weight at G0 (SD *n* = 13, HFD *n* = 11, HFD+MF = 11) and body weight at G15.5(SD *n* = 14, HFD *n* = 11, HFD + MF *n* = 11) in grams. (**b**) Weight gain during the gestation period from G0 until G15.5 in grams. SD *n* = 13, HFD *n* = 11, HFD + MF *n* = 11. (**c**) Body weight at G15.5, related to dams’ body length in grams/cm. SD *n* = 12, HFD *n* = 11, HFD + MF *n* = 8. (**d**) Litter sizes at G15.5. SD *n* = 14, HFD *n* = 11, HFD + MF *n* = 11. (**e**) PgWAT weight at G15.5, normalised to body weight in %. SD *n* = 13, HFD *n* = 8, HFD + MF *n* = 11. (**f**) Exemplary section from pgWAT of HFD group as used for fat cell size analysis with exemplified area (yellow) and diameter (blue). Scale bar equals 50 µm. (**g**) Mean area of pgWAT fat cells at G15.5 in µm^2^. SD *n* = 14, HFD *n* = 11, HFD + MF *n* = 11. (**h**) Mean diameter of pgWAT fat cells at G15.5 in µm. SD *n* = 14, HFD *n* = 11, HFD + MF *n* = 11. Due to missing data points (body weight at G0, body weight or pgWAT weight at G15.5, body length at G15.5), *n*-counts of mice included in analyses shown under a-i sometimes differ. SD: standard diet; HFD: high-fat diet; HFD + MF: high-fat diet plus metformin treatment; G: gestational day; pgWAT: perigonadal white adipose tissue; * *p*-value < 0.05; ** *p*-value < 0.01; *** *p*-value < 0.001; **** *p*-value < 0.0001.

**Figure 2 nutrients-14-02288-f002:**
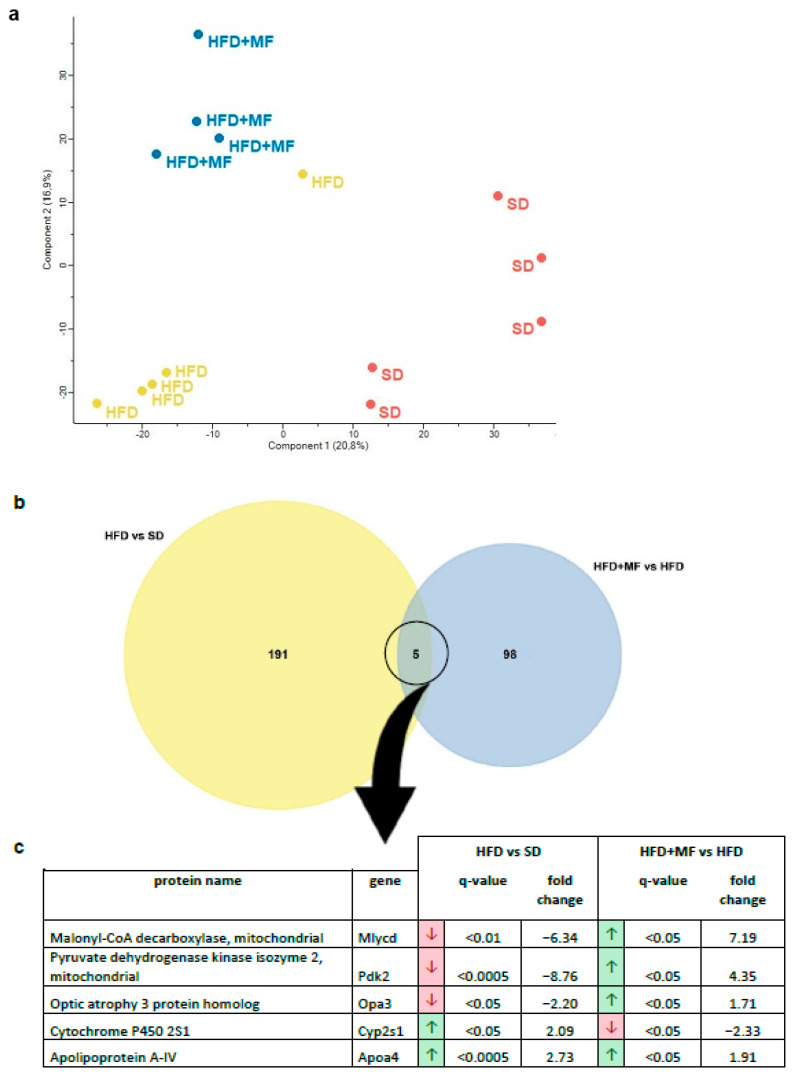
Proteins were extracted from pgWAT, detected by mass spectrometry and identified using MaxQuant software. Further data analysis was carried out with Perseus software. (**a**) A principal component analysis of all detected proteins was performed to visualise differential global protein expression. (**b**) A Venn diagram of significantly (fold change > 1.5, q-value < 0.05) and relevantly altered (full or 0 valid values per group) proteins to illustrate overlaps in comparison of HFD vs. SD (i.e., the number of proteins altered in pgWAT by HFD feeding) and HFD + MF vs. HFD (i.e., the number of proteins altered in pgWAT of HFD dams by metformin treatment during pregnancy). (**c**) The five overlapping proteins of both comparisons with their protein name, gene name, regulation, q-value and fold change. The arrows indicate either a downregulation (red) or upregulation (green) in obese (HFD) compared to lean dams (SD) (comparison 1—HFD vs. SD) or in metformin-treated HFD dams (HFD + MF) compared to obese dams without treatment (HFD) (comparison 2—HFD + MF vs. HFD). SD *n* = 5, HFD *n* = 5, HFD + MF *n* = 4. SD: standard diet; HFD: high-fat diet; HFD + MF: high-fat diet plus metformin treatment; pgWAT: perigonadal white adipose tissue.

**Figure 3 nutrients-14-02288-f003:**
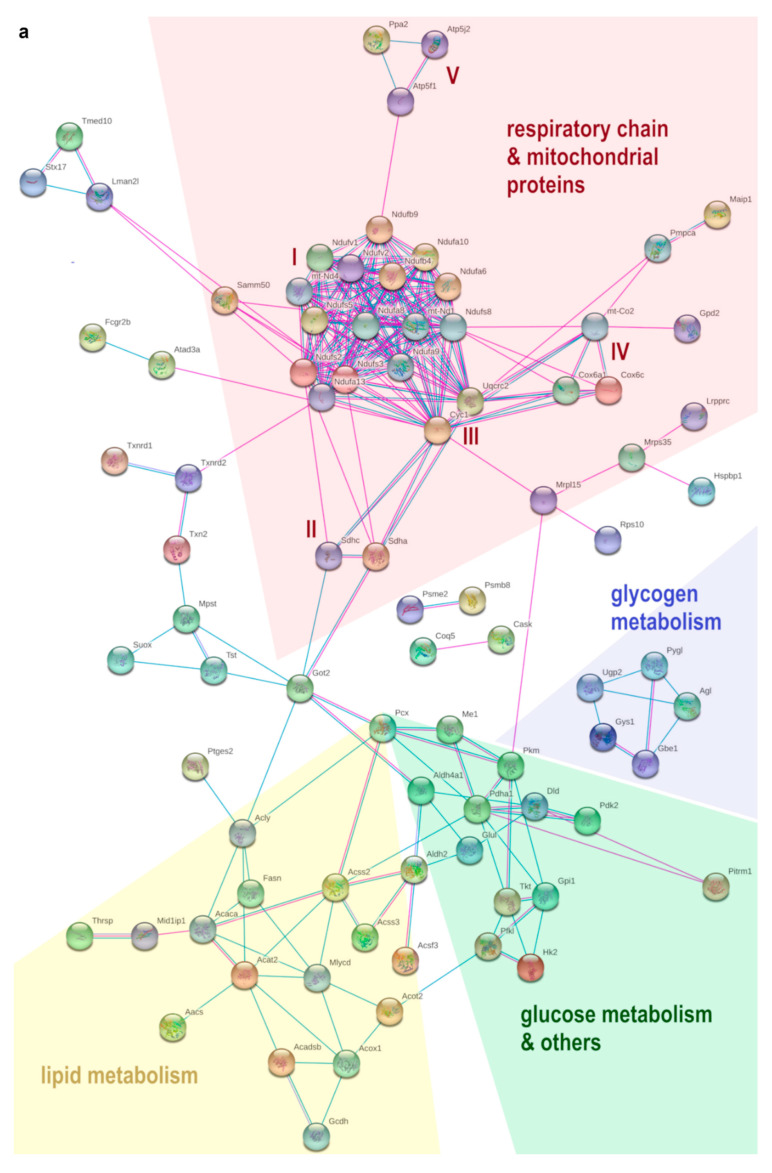
Potential interactions, indicated by lines between two proteins (blue: based on curated databases, pink: experimentally determined, purple: protein homology), were determined by STRING-analysis between significantly (fold change > 1.5, q-value < 0.05) and relevantly (full or 0 valid values per group) altered proteins in pgWAT proteome. Non-interacting proteins are hidden. (**a**) shows downregulated proteins and (**b**) shows upregulated proteins by HFD feeding compared to SD. Some functional enrichments of upregulated proteins are highlighted in colour: green: cellular component organisation, blue: cytoskeleton organisation. SD: standard diet; HFD: high-fat diet; pgWAT: perigonadal white adipose tissue.

**Figure 4 nutrients-14-02288-f004:**
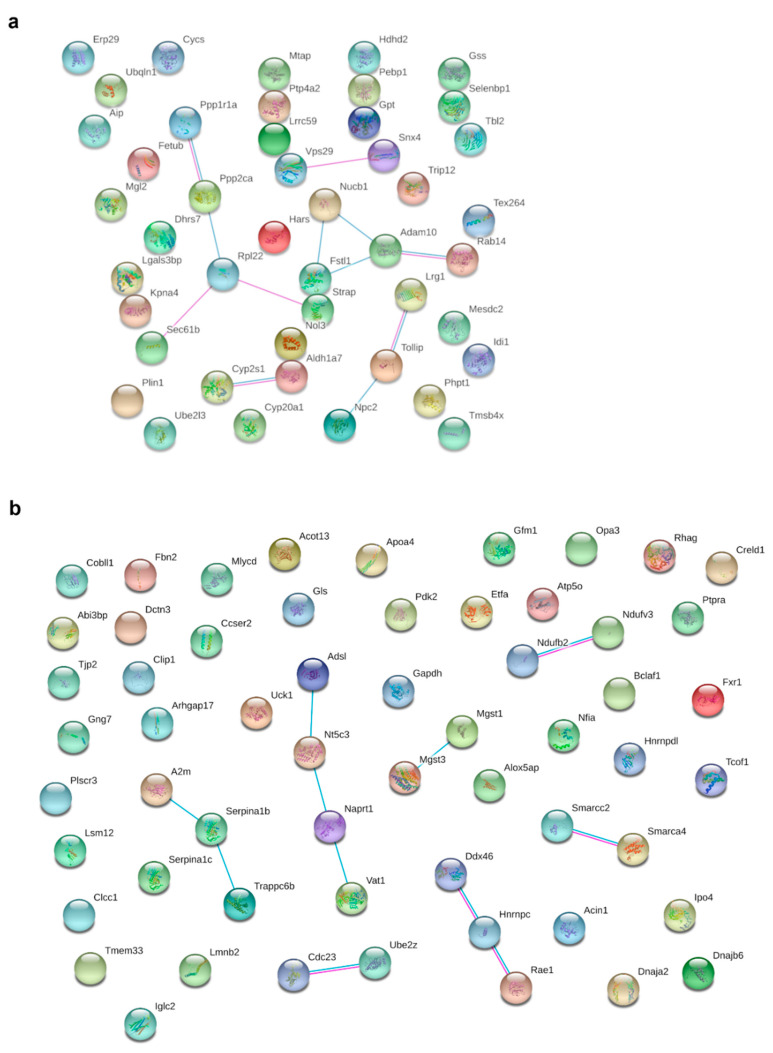
Potential interactions, indicated by lines between two proteins (blue: based on curated databases, pink: experimentally determined), were determined by STRING analysis between significantly (fold change > 1.5, q-value < 0.05) and relevantly (full or 0 valid values per group) altered proteins in pgWAT proteome. (**a**) shows downregulated proteins and (**b**) shows upregulated proteins by HFD + MF feeding compared to HFD feeding. HFD: high-fat diet; HFD + MF: high-fat diet plus metformin treatment; pgWAT: perigonadal white adipose tissue.

## Data Availability

Not applicable.

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
