# Peer review of "Metformin Prevents Key Mechanisms of Obesity-Related Complications in Visceral White Adipose Tissue of Obese Pregnant Mice"

_nutrients, 2022, doi:10.3390/nu14112288_

Round 1

Reviewer 1 Report

The study investigated that effects of Metformin on white adipose tissue of obese pregnant mice. The topic is of interest.

My main concern is that weight gain between SD and HFD group during gestation did not different, whilst HFD+Metformin group had significant gestational weight gain loss compared with SD and HFD. This may indicate adverse effects of metformin in this model. The timing of Metformin appears to be explained clearly and justified, e.g., soon after mating. Was this timing of administration particularly relevant to humans? Although the study focused on white adipose tissue, I would like to see how this means in the pregnant animals. Does this affect offspring growth trajectory?

The overall presentation of the paper could be benefit from major revisions.

  • There are some typos in the manuscript that requires spell checking.
  • Serum Metformin concentration. Authors commented the concentration detected in HFD+MF mice would be in the upper therapeutic range and cited one meta-analysis. Is there any relationship between dosage and serum concentration? Are the any guidelines for metformin concentration during pregnancy?
  • Body weight gain during pregnancy, Figure1 c showed body weight gain in bar chart. It would be more informative if the trend of weight gain throughout pregnancy using line graph.
  • Re Figure1 a-i: please explain why n are different in each group across graphs
  • Section 3.4 and figure2: Authors discussed commonly changed proteins between HFD vs SD, and between HFD vs HFD + Metformin (figure 2b). I am not quite followed what the 5 common proteins between HFD vs SD comparison and between HFD vs HFD + Metformin comparison have meanings in the study. Please discuss in terms of relevance to clinical setting? What the common five proteins have implications to Metformin use in pregnancy?
  • Figure 2c table, please clarify what group were based on increase/decrease of these proteins? E.g.,  Pdk2 decreased in HFD or SD compared with SD or HFD?    
  • Section 3.5 and figure 3&4: I don’t follow this section and the figure. Please clarify what these figures depicted. Why there are 3 x figure4?

Author Response

Thank you for your valuable time and your constructive feedback. We feel that the manuscript has improved after adressing your concerns. Listed below are the point by point responses to your comments.

The study investigated that effects of Metformin on white adipose tissue of obese pregnant mice. The topic is of interest.

My main concern is that weight gain between SD and HFD group during gestation did not different, whilst HFD+Metformin group had significant gestational weight gain loss compared with SD and HFD. This may indicate adverse effects of metformin in this model.

For human obese patients, weight gain during pregnancy is recommended to be lower than in normal weight pregnant women [1, 2]. Some even say that obese women should rather loose weight during pregnancy [3]. This however is controversially discussed [4]. We agree that strong loss in weight gain during pregnancy might be an issue and will discuss this problem in the manuscript (lines 411-415).

The timing of Metformin appears to be explained clearly and justified, e.g., soon after mating. Was this timing of administration particularly relevant to humans?

Metformin treatment starts either before and throughout pregnancy (PCOS – polycystic ovary syndrome or pre-existing diabetes mellitus type 2) or after detection of gestational diabetes mellitus (starting after first trimester) [5].

For lifestye interventions (increasing physical activity, changing eating habits) it is known that, when starting within or after first trimester, they have only little effect on e.g. maternal metabolic conditions. This is probably due to the fact that ‚early pregnancy metabolic conditions occur in the first trimester of pregnancy … prior to when most interventional trials are initiated‘ [6]. Hence, the idea for this project was to start the metformin intervention right after mating.

Since the timing of metformin treatment does not match exactly any current clinical scenario, we discuss this issue in more detail now in the manuscript (lines 67-71 and 77-80).

Although the study focused on white adipose tissue, I would like to see how this means in the pregnant animals. Does this affect offspring growth trajectory?

Data from this mouse model showed no effect of intrauterine metformin exposure on fetal body weight at G18.5 [7].  The analysis of offspring growth trajectory after birth is subject of a new ongoing project in our lab and we do not have any valid data sets so far. Other studies however indicate that children from women treated with metformin during pregnancy are heavier than children from untreated women [8-11] and animal studies imply the same [12, 13].

The overall presentation of the paper could be benefit from major revisions.

  • There are some typos in the manuscript that requires spell checking.

We carefully checked for typos and hopefully fixed all of them.

  • Serum Metformin concentration. Authors commented the concentration detected in HFD+MF mice would be in the upper therapeutic range and cited one meta-analysis. Is there any relationship between dosage and serum concentration? Are the any guidelines for metformin concentration during pregnancy?

Typically, daily metformin dosage during pregnancy ranges from 1000 (500 mg 1-0-1) to 2000 (1000 mg 1-0-1) mg/day [14]. Another review says that daily dosage during pregnancy ranges from 500 to 3000 mg metformin [15]. After dosage of 1000 mg metformin twice daily, plasma metformin concentration peaked at about 1500 ng/ml + 500 (mean + SD). Kinetics of metformin during pregnancy are also described in this publication, indeed demonstrating a relationship between dosage and plasma concentration of metformin [14].

As in our pregnant mouse model, median metformin concentration is 2.24 µg/ml, we’re above the therapeutic range in pregnant women. This will be discussed in the manuscript (lines 401-407) to let the reader know about this limitation.

  • Body weight gain during pregnancy, Figure1 c showed body weight gain in bar chart. It would be more informative if the trend of weight gain throughout pregnancy using line graph.

We have only 2 time points for body weight of the dams: G0 and G15.5. A line graph will look like this: see attached pdf file (line graph_gestational weight gain). If you prefer this version, we will be glad to replace the bar graphs with this line graph. Then, graph 1a and 1b have to be removed from the manuscript.

  • Re Figure1 a-i: please explain why n are different in each group across graphs

Thank you for pointing out to this issue/addressing this point!

In Fig. 1a, for SD we had animals included in the statistic that do not belong here (first, we had another group SD+MF included in the manuscript, that was deleted during the writing process due to missing clinical relevance). These animals were deleted from the statistics now, but results are the same.

In Fig. 1b-d, we excluded animals with less than 5 living pups to not falsify body weight data (this was mentioned in the manuscript line 192: „Dams with a litter size below five were excluded from the comparison of body weight at G15.5 and weight gain from G0 until G15.5.“). We now also performed a statistical analysis for all animals (animals with less than 5 living pups included) and got the same results as before. So to not confuse the reader, we now decided to go with the updated version of Fig. 1b-d, corrected the numbers in section 3.2 and deleted the corresponding text passage (lines 199-201) in the manuscript. The reason for all other variable n counts in different Fig. 1 sections is, that we sometimes miss single data points for animals. E.g., an animal or pgWAT was not weighted at G0 or G15.5, or body length of an animal was not measured at G15.5. We will clarify this in the manuscript now (see Figure legend 1).

  • Section 3.4 and figure2: Authors discussed commonly changed proteins between HFD vs SD, and between HFD vs HFD + Metformin (figure 2b). I am not quite followed what the 5 common proteins between HFD vs SD comparison and between HFD vs HFD + Metformin comparison have meanings in the study. Please discuss in terms of relevance to clinical setting? What the common five proteins have implications to Metformin use in pregnancy?

The five proteins Apoa4, Mlycd, Pdk2, Opa3 and Cyp2s1 have been of particular interest to us, as they were both regulated by HFD (compared to SD) and metformin intervention (HFD+MF compared to HFD). Especially a closer look at the last 4 proteins seemed important to us as they were counter-regulated in the two comparisons. Independent of what is known about these proteins for clinical settings, metformin treatment or obese pregnancy, it is exciting that metformin has the potential to sort of normalize their protein level back to control/SD levels. Hence, we can not add more information about these five proteins beyond what has been discussed in the manuscript already.

  • Figure 2c table, please clarify what group were based on increase/decrease of these proteins? E.g.,  Pdk2 decreased in HFD or SD compared with SD or HFD?   

The arrows in the table indicate if the protein was up/downregulated in HFD compared to SD (comparison 1) or in HFD+MF compared to HFD (comparison 2). We now added more information in the figure legend to better explain the table.

  • Section 3.5 and figure 3&4: I don’t follow this section and the figure. Please clarify what these figures depicted. Why there are 3 x figure4?

Figures 3 (comparison 1 – HFD compared to SD) and 4 (comparison 2 – HFD+MF compared to HFD) show potential protein protein interactions between significantly downregulated (a) or upregulated (b) proteins detected by STRING database. Potential interactions (based on experiments and curated databases) are indicated by lines between two proteins. We added this information and an explanation of the colour code of the lines in the figure legends.

There is only one Fig. 4 in our version of the manuscript. We are very sorry if there’s a format error causing multiplication of Fig. 4. However, as we do not see this issue in our version, we do not know how to fix the problem.

References

  1. Cedergren, M.I., Optimal gestational weight gain for body mass index categories. Obstet Gynecol, 2007. 110(4): p. 759-64.
  2. Artal, R., C.J. Lockwood, and H.L. Brown, Weight gain recommendations in pregnancy and the obesity epidemic. Obstet Gynecol, 2010. 115(1): p. 152-155.
  3. Bogaerts, A., et al., Weight loss in obese pregnant women and risk for adverse perinatal outcomes. Obstet Gynecol, 2015. 125(3): p. 566-575.
  4. Catalano, P.M., et al., Inadequate weight gain in overweight and obese pregnant women: what is the effect on fetal growth? Am J Obstet Gynecol, 2014. 211(2): p. 137.e1-7.
  5. Nguyen, L., S.Y. Chan, and A.K.K. Teo, Metformin from mother to unborn child - Are there unwarranted effects? EBioMedicine, 2018. 35: p. 394-404.
  6. Catalano, P. and S.H. deMouzon, Maternal obesity and metabolic risk to the offspring: why lifestyle interventions may have not achieved the desired outcomes. Int J Obes (Lond), 2015. 39(4): p. 642-9.
  7. Nusken, E., et al., Maternal High Fat Diet and in-Utero Metformin Exposure Significantly Impact upon the Fetal Renal Proteome of Male Mice. J Clin Med, 2019. 8(5).
  8. Carlsen, S.M., M.P. Martinussen, and E. Vanky, Metformin's effect on first-year weight gain: a follow-up study. Pediatrics, 2012. 130(5): p. e1222-6.
  9. Ijas, H., et al., A follow-up of a randomised study of metformin and insulin in gestational diabetes mellitus: growth and development of the children at the age of 18 months. BJOG, 2015. 122(7): p. 994-1000.
  10. Hanem, L.G.E., et al., Metformin Use in PCOS Pregnancies Increases the Risk of Offspring Overweight at 4 Years of Age: Follow-Up of Two RCTs. J Clin Endocrinol Metab, 2018. 103(4): p. 1612-1621.
  11. Rowan, J.A., et al., Metformin in gestational diabetes: the offspring follow-up (MiG TOFU): body composition at 2 years of age. Diabetes Care, 2011. 34(10): p. 2279-84.
  12. Schoonejans, J.M., et al., Maternal Metformin Intervention during Obese Glucose-Intolerant Pregnancy Affects Adiposity in Young Adult Mouse Offspring in a Sex-Specific Manner. Int J Mol Sci, 2021. 22(15).
  13. Salomaki, H., et al., Prenatal metformin exposure in mice programs the metabolic phenotype of the offspring during a high fat diet at adulthood. PLoS One, 2013. 8(2): p. e56594.
  14. Liao, M.Z., et al., Effects of Pregnancy on the Pharmacokinetics of Metformin. Drug Metab Dispos, 2020. 48(4): p. 264-271.
  15. Butalia, S., et al., Short- and long-term outcomes of metformin compared with insulin alone in pregnancy: a systematic review and meta-analysis. Diabet Med, 2017. 34(1): p. 27-36.

Reviewer 2 Report

I appreciate the opportunity to review this interesting paper on the action of metformin on  visceral white adipose tissue of obese pregnant mice.

Globally, the number of pregnancies impacted by obesity has grown significantly in recent years. Pregnancy weight gain is related to an increase in complications for both the mother and the newborn, and research suggests that maternal obesity may have a long-term influence on the cardiovascular health and cognition of offspring. While the effects of maternal obesity are complex, the higher glucose levels reported in obese non-diabetic women compared to normal weight women may impact the outcome of pregnancy. In many overweight and obese pregnant women studies, metformin has been tested to determine whether it can help to reduce the risk of pregnancy-related morbidity and mortality.

As the authors have pointed out, little research has been done on the effects of metformin on maternal WAT in obese pregnancies. In order to learn more about this issue, researchers used a mouse model of maternal obesity, in which therapy with metformin was used from the beginning of pregnancy.

Some of the paper's strongest points are that it tackles an intriguing and topical subject, that it is well-written, that the experiments were well-conducted, and that the analysis was well-done.

Considering these strengths, though, as I read the manuscript, I found some areas in which I would have appreciated greater clarity. I believe the paper could be further strengthened by added information about:

  • A flow chart showing the experimental procedure should be included as it makes the experimental setting more visible to the reader.
  • The authors do not provide the composition of diets but refer the reader to previous publications. I would suggest including this information in the Supplementary Material.
  • I doubt whether they were right to use the perigonadal WAT (pgWAT). A better solution would be to use depots that drain their blood into the portal vein (i.e. omental and mesenteric), which can be of great importance in metabolic disorders.,. Of course, omental is often non-existent in young mice and relatively small in older animals, but mesenteric seems to be a better choice.I agree that the perigonadal fat pads are typically the largest and most readily accessible fat pads in mice, and as a result, they are the most frequently used in the literature. There are no human parallels to them, however.

Author Response

Thank you for your valuable time and your constructive feedback. We feel that the manuscript has improved after adressing your concerns. Listed below are the point by point responses to your comments.

I appreciate the opportunity to review this interesting paper on the action of metformin on  visceral white adipose tissue of obese pregnant mice.

Globally, the number of pregnancies impacted by obesity has grown significantly in recent years. Pregnancy weight gain is related to an increase in complications for both the mother and the newborn, and research suggests that maternal obesity may have a long-term influence on the cardiovascular health and cognition of offspring. While the effects of maternal obesity are complex, the higher glucose levels reported in obese non-diabetic women compared to normal weight women may impact the outcome of pregnancy. In many overweight and obese pregnant women studies, metformin has been tested to determine whether it can help to reduce the risk of pregnancy-related morbidity and mortality.

As the authors have pointed out, little research has been done on the effects of metformin on maternal WAT in obese pregnancies. In order to learn more about this issue, researchers used a mouse model of maternal obesity, in which therapy with metformin was used from the beginning of pregnancy.

Some of the paper's strongest points are that it tackles an intriguing and topical subject, that it is well-written, that the experiments were well-conducted, and that the analysis was well-done.

Considering these strengths, though, as I read the manuscript, I found some areas in which I would have appreciated greater clarity. I believe the paper could be further strengthened by added information about:

  • A flow chart showing the experimental procedure should be included as it makes the experimental setting more visible to the reader.

This is an important point and we added a flow chart about the experimental setup of the mouse groups to the Supplementary Material (Figure S1).

  • The authors do not provide the composition of diets but refer the reader to previous publications. I would suggest including this information in the Supplementary Material.

We added detailed information about the diets used in the study in the Supplementary Material (Table S1).

  • I doubt whether they were right to use the perigonadal WAT (pgWAT). A better solution would be to use depots that drain their blood into the portal vein (i.e. omental and mesenteric), which can be of great importance in metabolic disorders.,. Of course, omental is often non-existent in young mice and relatively small in older animals, but mesenteric seems to be a better choice.I agree that the perigonadal fat pads are typically the largest and most readily accessible fat pads in mice, and as a result, they are the most frequently used in the literature. There are no human parallels to them, however.

Thank you for raising this issue. Unfortunately, we completed all mouse experiments by now and it is therefore not possible to dissect any other fat pads. But we added a section into the limitations part of the discussion to let the reader know that the choice of the analysed fat pad was not optimal (lines 516-519).